# CHMM Object Detection Based on Polygon Contour Features by PSM

**DOI:** 10.3390/s22176556

**Published:** 2022-08-30

**Authors:** Shufang Zhuo, Yanwei Huang

**Affiliations:** 1College of Automation Engineering, Fujian Polytechnic of Information Technology, Fuzhou 350003, China; 2College of Electrical Engineering and Automation, Fuzhou University, Fuzhou 350116, China

**Keywords:** object detection, contour, polygonal approximation, piecewise split–merge algorithm, Coupled Hidden Markov Model

## Abstract

Since the conventional split–merge algorithm is sensitive to the object scale variance and splitting starting point, a piecewise split–merge polygon-approximation method is proposed to extract the object contour features. Specifically, the contour corner is used as the starting point for the contour piecewise approximation to reduce the sensitivity of the contour segment for the starting point; then, the split–merge algorithm is used to implement the polygon approximation for each contour segment. Both the distance ratio and the arc length ratio instead of the distance error are used as the iterative stop condition to improve the robustness to the object scale variance. Both the angle and length as two features describe the shape of the contour polygon; they have a strong coupling relationship since they affect each other along the contour order relationship. To improve the description correction of the contour, these two features are combined to construct a Coupled Hidden Markov Model to detect the object by calculating the probability of the contour feature. The proposed algorithm is validated on ETHZ Shape Classes and INRIA Horses standard datasets. Compared with other contour-based object-detection algorithms, the proposed algorithm reduces the feature number and improves the object-detection rate.

## 1. Introduction

Object detection is one of the hotspots and difficulties in computer vision research, and is the basic technology of high-level vision tasks [1]. Compared with the object-detection algorithm based on deep learning, the sample size and computation complexity of the contour-based object-detection algorithm are relatively small and require relatively low hardware specifications due to the reduced hardware resource consumption. Therefore, the relevant research has been receiving significant attention in recent years [2,3,4,5,6,7]. The detection algorithm uses shape as a feature to accurately detect and identify objects and is applied to the visual system of indoor mobile robots [8]. However, there are still many challenges in the contour-detection algorithm. The performance of the detection algorithm is susceptible to noise, feature dimensions, and feature descriptors. The robust Hausdorff distance is used to evaluate the similarity between the model feature point set and the object feature point set [9]. However, it is sensitive to noise and only suitable for objects with smooth surface boundaries. In order to improve the robustness to noise, the contour shape is segmented, and each contour segment is matched by constellation shape analysis to improve the feature robustness in the clutter image [10]. However, the computational complexity is increased greatly. The contour edges are fragmented to locally match the object contour model in ergodicity [11]. This method improved the object-detection correction under clutter background or occlusion conditions. These methods improve the anti-noise robustness from point set feature matching to segment feature matching. The Hough transform is used to extract the straight line segment of the warship contour to calculate the turn angle among adjacent straight line segments [12]. The angles are used as features to establish a Hidden Markov Model for object detection. This method takes the angle as a contour feature to detect an image with a simple background. In order to enhance the feature expression for the object contour, the angle, distance and arc length chord ratio are used as the contour descriptor for local matching to improve the object-detection accuracy in the clutter background image [6]. Therefore, the technique combining straight line segments with angles is a more efficient method for contour-based object detection. In image processing, the polygon-approximation method is often used to replace the curve with multiple line segments to simplify the contour model, and still retain the key information of the contour. The contour feature of the polygon approximation is more robust to image noise than the point set feature in [9], and has a higher feature matching efficiency. The polygon-approximation method for the contour feature is proposed to extract two variables as the angle and the length to improve the anti-interference ability and reduce the contour model complexity.

Since the split–merge algorithm (SM) can retain the basic contour features well, it is widely used in the industrial production field [13]. SM is sensitive to both the scale variance of the object and the starting point of the split. A modified piecewise split–merge (PSM) polygon-approximation algorithm is proposed to improve the SM robustness for the object scale variance and starting point. Specifically, the sharpness is used to detect the corners of object contours as the starting points [14]. The corner sequences are used to segment the object contour as piecewise contours. Each contour segment is approximated by the SM algorithm to extract the features of the angle and length. Both the distance ratio and the arc length ratio instead of the distance error are used as the iterative stop conditions for SM to improve the robustness to the object scale variance.

In order to improve the matching efficiency of the contour-based object-detection algorithm, the Hidden Markov Model is introduced into the classification process [12,15,16,17] to achieve good performance. The Coupled Hidden Markov Model (CHMM) is used to describe the existence of multiple probabilistic models of interrelated random processes [18]. CHMM not only has the characteristics of the Hidden Markov Model (HMM), but can also describe the interdependence between associated stochastic processes, and it is already applied for fault detection [19]. Considering the correlation between the length of the polygon segment and the angle, PSM-based CHMM (PSM-CHMM) is introduced to detect objects. PSM is used to extract the feature variables of the angle and length for object detection and CHMM is used as the object detector with the two features to increase the detection accuracy.

## 2. Polygon Contour Feature Extraction

The global probability boundary algorithm (gPb) is used to segment the image and extract the object contour [20]. The gPb algorithm considers the image color, grayscale, texture and other information to extract high-quality image contours. Figure 1 is the contour of the cup image extracted by the gPb algorithm. The object contours with different details can be obtained with different thresholds *k*. Polygon approximation is used to extract the contour features.

The PSM polygon-approximation algorithm is proposed to extract contour features. Firstly, the corner-detection algorithm based on sharpness is used to determine the contour corners, and the contour is segmented by the corner, which reduces the dependence of the contour segment on the starting point. Secondly, the PSM algorithm is used to approximate each contour segment. The distance ratio and arc length ratio are used to replace the distance error as the iterative stop condition, which improves the robustness of the object scale variance, such as the size, rotation, etc. Finally, both the angle and length extracted for the approximate polygon as two features describe the contour shape.

### 2.1. Contour Piecewise by Sharpness-Based Corner

SM usually performs polygon approximation on the entire contour, and can reserve the key feature information of the contour well. However, starting points would not remain invariable when the object contour is affected by the distance, such as image noise, object rotation, occlusion, and scale variance. The different starting points will result in different SM results. In order to reduce the difference of contour-approximation results, the authors of [10] proposed a method that extracts features after the segmentation of contours, thus improving the robustness of the approach when applied on images representing cluttered scenes. The starting point is the key technology for contour segmentation. A method to detect the corner is proposed by calculating the sharpness of each point on the contour line with non-maximal suppression [14]. Based on the sharpness of the contour, a contour-segmentation method is proposed by detecting the contour corners as starting points of contour segments. This method to detect corners using sharpness has strong anti-interference ability, small calculation time, and accurate positioning. Every contour segment is estimated by the polygon approximation with SM to extract the features.

Let $p_{i}$ denote the pixel points on the contour in Figure 2. Considering $p_{i}$ as the center point of a given contour segment, points $p_{i\;-\;1}$, …, $p_{i\;-\;r}$ lay on its left and points $p_{i\;+\;1}$, …, $p_{i\;+\;r}$ on its right, thus composing the whole contour. These 2r + 1 points are the support region of the point $p_{i}$. In Figure 2, the line segments $p_{i}p_{i\;-\;r}$, $p_{i}p_{i\;+\;r}$ are called “support arms”, and α is the “support angle” of the “support area”. Since the support region is a very small area, the support angle α can be approximated:(1)$$\sin\left(\alpha /2\right)\;=\;\frac{|p_{i\;-\;r}p_{i\;+\;r}|}{|p_{i}p_{i\;-\;r}\left|\;+\;\right|p_{i}p_{i\;+\;r}|}$$

The sharpness $\eta_{i}$ of the point $p_{i}$ is defined as:(2)$$\eta_{i}\;=\;1\;-\;\frac{|p_{i\;-\;r}p_{i\;+\;r}|}{|p_{i}p_{i\;-\;r}\left|\;+\;\right|p_{i}p_{i\;+\;r}|}$$

The larger the $\eta_{i}$, the sharper the angle. The sharpness $\eta_{i}$ is calculated for each point on the contour. When $\eta_{i}>T$, the point is marked as a candidate corner with a set threshold *T*.

The sharpness of the contour is calculated as the maximum sharpness of the points in Equation (Equation 2):(3)$${\bar{\eta }}_{i}\;=\;\max_{\left|\left.j\;-\;i\right|\leq r\right.}\eta_{j}$$
where j∈[i − r,i + r]. The higher the sharpness, the better the robustness of the point. The parameter *r* and the threshold *T* are selected by the bending degree of the contour. Parameter *r* is generally set to 3~15, and *T* is generally set to 0.05~0.15. Here, *T* and *r* are selected as T = 0.05, r = 13. The corner is detected as the starting point of the contour polygon segment approximation. Figure 3 is the contour segmentation by the corner for the object bat in the MPEG-7 data set [21]. Figure 4 is the bat divided into two parts to simulate different degrees of occlusion. Both Figure 3 and Figure 4 show that the segmentation of the remaining contours is affected by neither the contour completeness nor incompleteness.

### 2.2. PSM Polygonal-Approximation Algorithm

When the contours are segmented into segments by corners, each contour segment is estimated by the SM polygon-approximation algorithm. The variance of the contour starting point does not affect the subsequent contour segments for the polygon approximation compared with the approximation method of the entire contour. However, the distance threshold is used as the iterative stop condition for the SM algorithm, which results in a different number of vertices for polygons when the object scale varies. In order to improve the robustness of scale variance for the SM algorithm, the distance ratio and the arc length ratio as the stopping condition instead of the distance threshold are implemented into the PSM algorithm. Figure 5 is the process of the PSM polygon-approximation algorithm for the bat contour from pixel point A to pixel point B. The PSM polygon-approximation algorithm obtains four pairs of line segments as AD, DC, CE and EB.

Figure 6 is the basic principle of the PSM polygon-approximation algorithm. Let $p_{1},p_{2},\;\ldots ,p_{n}$ be the sequence of pixel points for a given contour; the point $p_{1}\;=\;\left(x_{1},y_{1}\right)$ is the starting point A; pixel $p_{n}\;=\;\left(x_{n},y_{n}\right)$ is the ending point B. The two points A and B are connected as the initial approximation line segment. The length of line segment AB is $d_{0}$, the arc length of contour AB is $l_{0}$, the distance $d_{i}$ of the point $p_{i}$ to the line segment AB is calculated in turn to select the pixel point C corresponding to the maximum distance $d_{max}$. It can obtain the length $l_{c}$ of the AC arc. The distance ratio $d_{max}/d_{0}$ and the arc length ratio $l_{c}/l_{0}$ are used as the stopping conditions for the PSM polygonal-approximation algorithm. If it satisfies the stopping condition,
(4)$$d_{max}/d_{0}<\varepsilon_{1}\cup l_{c}/l_{0}<\varepsilon_{2}$$
the split stops. If it satisfies the condition,
(5)$$d_{max}/d_{0}>\varepsilon_{1}\cap l_{c}/l_{0}>\varepsilon_{2}$$
the split continues. The pixel points A and C are considered as the starting point and the end point, and the pixel points B and C are considered as the starting point and the ending point, respectively. The splitting process is performed again until the stopping condition satisfies Equation (Equation 4).

The SM algorithm uses the distance as the stopping condition. The distance is an absolute physical quantity. When the contour scale varies, the threshold needs to be adjusted to obtain the approximation result of the same number of vertices. The ratio stop condition of the PSM algorithm is a relative physical quantity, and the number of polygon vertices will remain consistent even if the object scale varies. Therefore, the analysis can be conducted in two different cases.

Case 1: polygon vertex number comparison for completed contour. PSM and SM (Ramer et al. [13]) plural are applied to approximate the bat contour, respectively. The bats are included in the MPEG-7 data set. Figure 7 shows bat images at different scales (100, 80, and 60) and rotations ($90^{\circ }$ and $180^{\circ }$ clockwise). Figure 8 and Figure 9 compare the approximation results of the SM algorithm and PSM algorithm. Table 1 shows the number of polygon vertices.

In Table 1, the feature number by the PSM algorithm is much smaller than that by SM algorithm. When the object scale varies and rotates, the number of vertices increases gradually under the same threshold for the SM algorithm and remains almost invariant for the PSM algorithm. Compared to SM, PSM improves the robustness of scale variance for the ratio of distance ratio and arc length as the stopping conditions.

Case 2: the location comparisons of polygon vertices for incomplete contour by PSM. In Figure 10, the bat object is divided into two parts. They are fitted by the polygon approximation with PSM, respectively. The location of polygon vertices in each part is consistent with the approximation of the complete contour. So, the PSM algorithm cannot affect the location of polygon vertices for either the complete contour or not.

### 2.3. Contour Feature

After the contour polygon is obtained by the PSM algorithm, polygon feature variables are extracted as contour descriptors. In [12], the angle of the line segment is used as the contour feature of the warship for object detection in the satellite image. The length and angle of the line segment are the two major elements that determine the shape of the polygon. These two feature variables determine the unique shape of the polygon, and the direction of the line segment. However, the changes of the two variables are random and asynchronous. In Figure 11, the polygon consists of a set of directed line segments with the angle and length. They are the two necessary factors to describe a directed line segment. Therefore, the polygon contour feature variable is determined as: the angle between the line segment length and the adjacent line segment. Let $P_{1},P_{2},\;\ldots ,P_{k},\;\ldots ,P_{n}$ be the ordered polygon vertices for a given contour *S*, *n* is the number of the polygon vertices. $\vec{a}\;=\;\vec{P_{k\;-\;1}P_{k}}$ and $\vec{b}\;=\;\vec{P_{k}P_{k\;+\;1}}$ indicate two adjacent directed line segments. The angle between the adjacent line segments $\theta_{k}$ is calculated by
(6)$$\theta_{k}\;=\;sign\left(\vec{a}\times \vec{b}\right)\arccos\left(\frac{\vec{a}\cdot \vec{b}}{|\vec{a}\left|\cdot \right|\vec{b}|}\right)$$

The length of the directed segment is $L_{k},k\;=\;1,2,\;\ldots ,n$. The length and the angle are normalized to improve the robustness to the scale variance. The descriptor $D_{S}$ of the contour *S* is
(7)$$D_{s}\;=\;\left[\begin{array}{ccc} \theta_{1} & \ldots & \theta_{n}\\ L_{1} & \ldots & L_{n} \end{array}\right]$$

In order to verify the consistency of the length and angle feature extracted by the PSM polygon-approximation algorithm, Figure 12 shows the two feature quantity curves along the contour. The length $L_{k}$ and the angle $\theta_{k}$ sequences are extracted by the PSM algorithm and the SM algorithm for the bat object with the three scales of 100%, 60%, and 90°, respectively. The threshold of the SM algorithm is 4. The thresholds of the PSM algorithm are $\varepsilon_{1}\;=\;0.03$ and $\varepsilon_{2}\;=\;0.03$ in Equation (Equation 4).

Figure 12a,c are the curves of the length and angle sequences extracted by the PSM algorithm for the bat with the three scales of 100%, 60%, and $90^{\circ }$. Figure 12b,d are the curves extracted by the SM algorithm. The three curves for the bat with the three scales of 100%, 60%, and 90° are much more consistent in Figure 12a than those in Figure 12b. The same is the case for Figure 12b,d. Moreover, the trends of the two features extracted on the same contour are asynchronous. So the two features of angle and length would describe the contour polygon more correctly. In the simulations, the computer CPU is AMD Ryzen7 4899H 2.90 GHz and 8 G memory with Matlab R2016a, the running times are 0.12 s for the PSM algorithm, and 0.18 s for the SM algorithm.

## 3. Experiment and Result Analysis

### 3.1. CHMM

HMM has been applied in object detection, and shows good rapidity and robustness, especially in contour-based object-detection research [12,15,16,17]. HMM is typically trained with a single feature of the object contour. Since the structure of HMM is a single-state Bayesian network, HMM is not suitable for multi-feature models. As contour descriptors, the length and the angle are used to determine the unique shape of the polygon. They have strong coupling relationship, but their trends on the same contour are different and random. For the feature coupling, CHMM with two contour features is constructed to improve the accuracy of object detection and anti-noise robustness.

CHMM has two processes, one is the training process, and the other is the testing process. For the training process, the two features of angle and length are extracted by the PSM algorithm for the training image. CHMM is trained with the two features by the EM algorithm proposed in [22] to obtain the classifier set $E\;=\;\lambda_{i}$; *i* is the number of categories. The larger the number of hidden states, the more accurate the model, but the training efficiency will be lower. In practice, the number of hidden states is four. In the process of parameter revaluation, the EM algorithm is iterated continuously to increase P(O|λ) until $|log(P_{j\;+\;1}\left(O|\lambda \right)\;-\;P_{j}\left(O|\lambda \right))|<\tau$, *j* is the iteration number. For the testing process, the PSM algorithm extracts the contour polygon feature sequence for the object contour of the test image. The forward–backward algorithm is used to calculate the probability $P(O|\lambda_{i});i$ is the object category number; *O* is the contour feature sequence. The object is identified for the given category by the rule of maximal probability, which must be larger than the given threshold.

### 3.2. Experimental Results

In the experiment, the standard datasets ETHZ Shape Classes [23] and INRIA Horses [24] were selected to verify the performance of PSM-CHMM. ETHZ Shape Classes contains 255 images of five categories of objects, including 40 apple logos, 48 bottles, 87 giraffes, 32 swans, and 48 mugs. The half images for every category as the training set are used to extract the contour feature sequence. The other half images are used as test sets. The INRIA Horses Dataset contains horse images in different scenes, including 170 images with horses and 170 images without horses. Fifty images with horses are used as the training set; the other images are used as the testing set. For the feature extraction process, the PSM threshold is $\varepsilon_{1}\;=\;\varepsilon_{2}\;=\;0.03$. For the training process, The CHMM parameters are estimated by the EM algorithm. P(O|λ) is considered to converge to the maximal value, when the P(O|λ) increment of the estimated model parameters for two consecutive times satisfies $|log(P_{j\;+\;1}\left(O|\lambda \right)\;-\;P_{j}\left(O|\lambda \right))|<0.001$ to stop the training process. In the experiment, the single-chain θ-HMM and the *L*-HMM were trained with only one angle and length feature for object detection, respectively. Bounding box is the smallest external rectangle of the contour. The performance of PSM-CHMM also compares with the two contour-based methods by Ferrari [1,7]. The indexes of Detection Rate/False-Positive (DR/FPPI) per image are used to verify the performance. Figure 13 is the DR/FPPI curves for the six categories of objects in the ETHZ Shape Classes and INRIA Horses data sets. Table 2 shows the detection rates for 0.3/0.4 FPPI on ETHZ shape classes and INRIA Horses.

In Figure 13, the yellow line and the purple line present DR/FPPI by the single-chain θ-HMM and *L*-HMM, respectively. Their DR/FPPI are much less than that with PSM-CHMM. CHMM can improve the detection accuracy since CHMM considers the feature coupling effects. Table 2 shows the DR comparisons. The index of DR is much higher than the other three methods. So, PSM-CHMM improves the accuracy of object detection. Figure 14 presents some examples of detection results for the apple logo, giraffe, bottle, swan, and horse, where the contour polygon model is given in first column, the second column is the test images, the third column is the contour extraction results, and the fourth column is the detection results.

## 4. Conclusions

Since the SM algorithm is sensitive to the object scale variance and splitting starting point, the PSM polygon-approximation algorithm is proposed to extract the object contour features. The PSM algorithm has some advantages. The contour corner is used as the starting point for the contour piecewise approximation to reduce the sensitivity of the contour segment on the starting point, and both the distance ratio and the arc length ratio instead of the distance error are used as the iterative stop condition to improve the robustness to the object scale variance. The PSM algorithm is applied to a bat image included in the MPEG-7 data set to verify the performance. In different scales and rotated experiments, the indexes of the polygon vertices are almost invariant for the PSM algorithm. The length and angle of the line segment obtained with the PSM algorithm are the two major elements that determine the shape of the contour polygon. The two features have a strong coupling relationship, but the changes of the two variables are random and asynchronous. CHMM with two contour features is constructed to improve the accuracy of object detection and anti-noise robustness. The ETHZ Shape Classes and INRIA Horses datasets are selected to verify the performance for PSM-CHMM in the experiment. The indexes of DR/FPPI and DR for 0.3/0.4 FPPI indicate that PSM-CHMM has a much better performance than the other three methods.

## Figures and Tables

**Figure 1 sensors-22-06556-f001:**
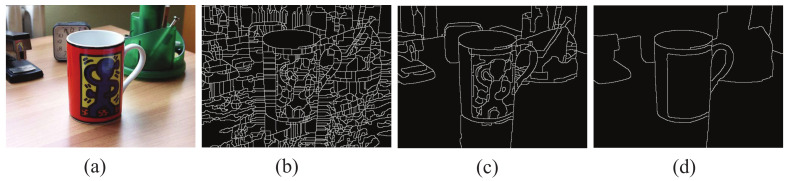
Contour of the cup image extracted by gPb with different thresholds. (**a**) Original image, (**b**) *k* = 0.01, (**c**) *k* = 0.3, (**d**) *k* = 0.6.

**Figure 2 sensors-22-06556-f002:**
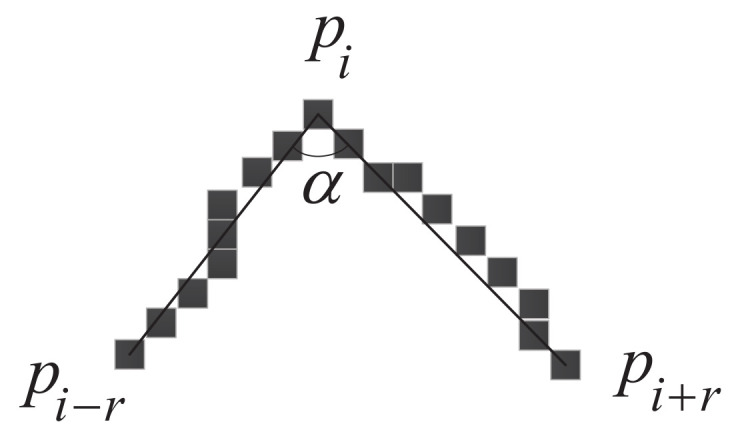
The support angle of point on contour.

**Figure 3 sensors-22-06556-f003:**
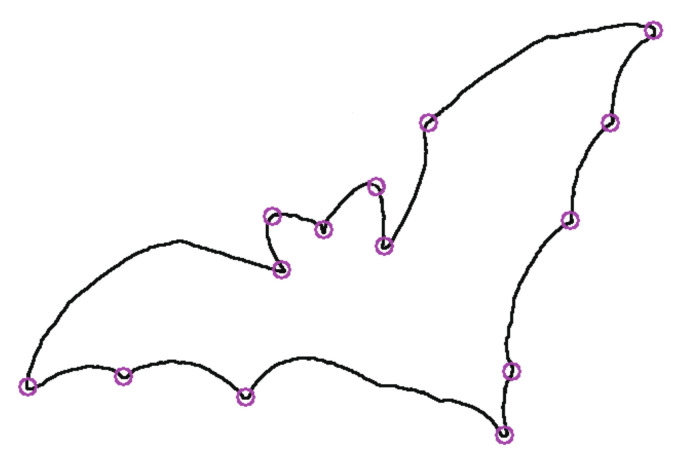
Contour segmentation by corner.

**Figure 4 sensors-22-06556-f004:**
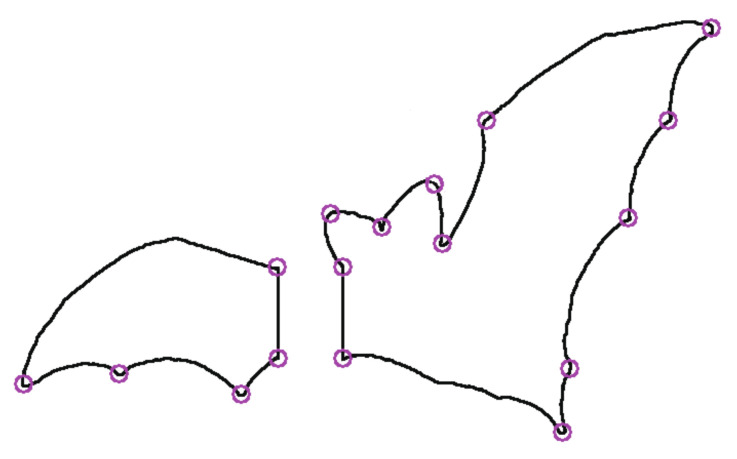
Incomplete contour segmentation.

**Figure 5 sensors-22-06556-f005:**
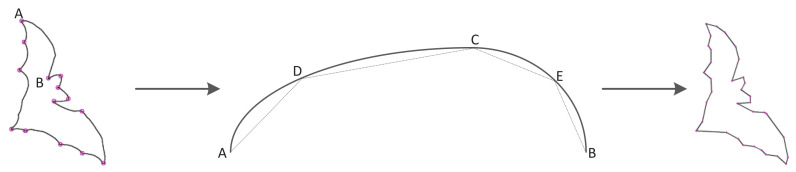
PSM polygon approximation with the points A, D, C, E, B.

**Figure 6 sensors-22-06556-f006:**
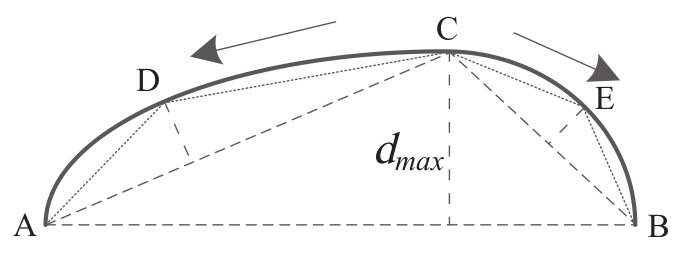
The basic principle of PSM polygon approximation.

**Figure 7 sensors-22-06556-f007:**
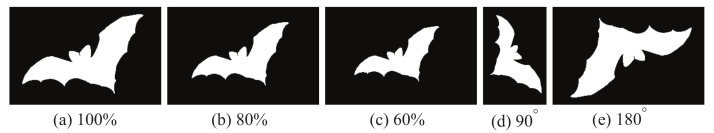
Original image at full scale (**a**), 80% scale (**b**), 60% scale (**c**), $90^{\circ }$ rotations (**d**), $180^{\circ }$ rotations (**e**).

**Figure 8 sensors-22-06556-f008:**
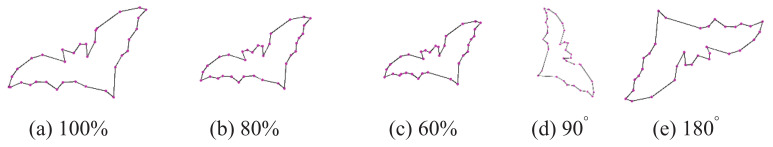
Approximation results of PSM algorithm on (**a**) original image at full scale, (**b**) original image at 80% scale, (**c**) original image at 60% scale, (**d**) original image at $90^{\circ }$ clockwise rotations, (**e**) original image at $180^{\circ }$ clockwise rotations.

**Figure 9 sensors-22-06556-f009:**
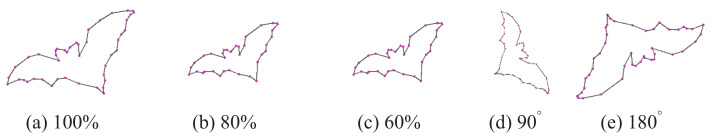
Approximation results of SM algorithm on (**a**) original image at full scale, (**b**) original image at 80% scale, (**c**) original image at 60% scale, (**d**) original image at $90^{\circ }$ clockwise rotations, (**e**) original image at $180^{\circ }$ clockwise rotations.

**Figure 10 sensors-22-06556-f010:**
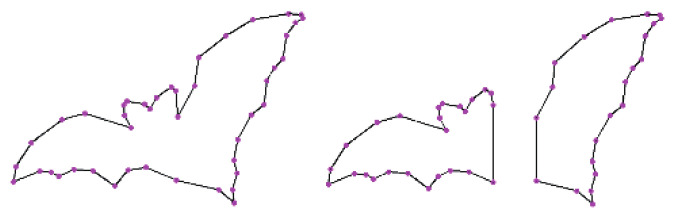
Incomplete contour-approximation result.

**Figure 11 sensors-22-06556-f011:**
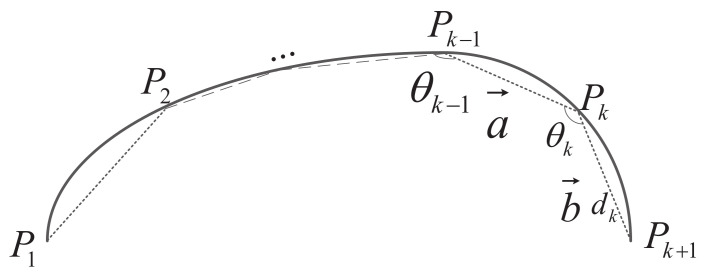
Polygon of the angle and length.

**Figure 12 sensors-22-06556-f012:**
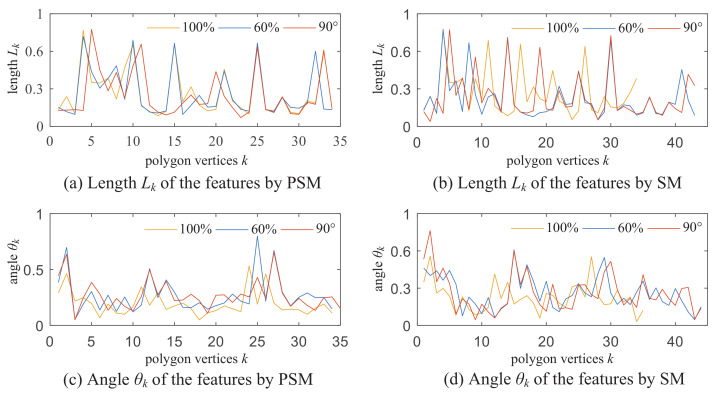
Features of the angle and length.

**Figure 13 sensors-22-06556-f013:**
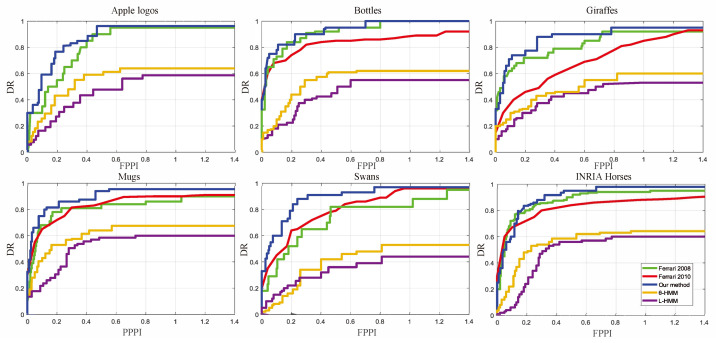
Comparison of DR/FPPI curves on ETHZ shape classes and INRIA Horses.

**Figure 14 sensors-22-06556-f014:**
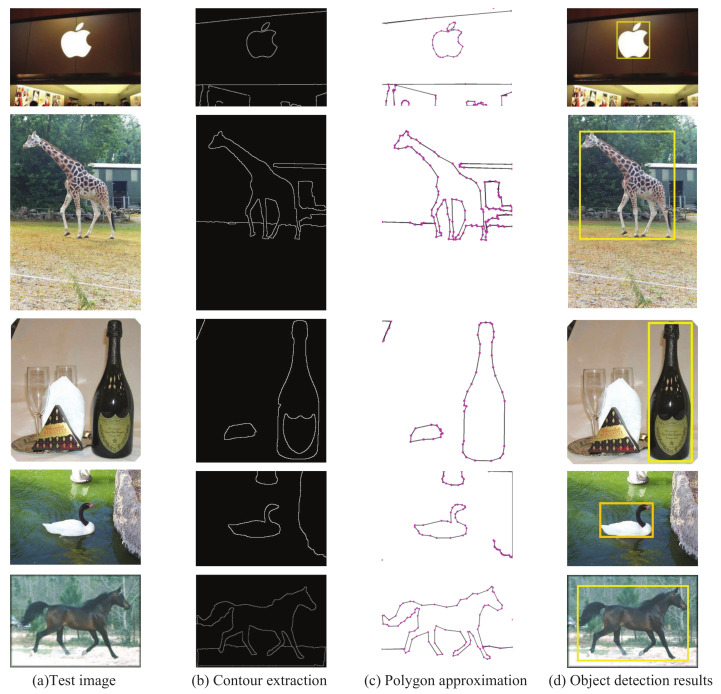
Some examples of detection results.

**Table 1 sensors-22-06556-t001:** Comparison of the number of polygon vertices.

Algorithm	Threshold	100%	80%	60%	$90^{\circ }$	$180^{\circ }$
SM	2	76	65	58	74	71
SM	4	45	40	36	45	45
PSM	0.03, 0.03	36	35	35	35	34
PSM	0.05, 0.05	26	26	26	26	26

**Table 2 sensors-22-06556-t002:** DR comparisons when FPPI = 0.3/0.4.

	θ-HMM	*L*-HMM	Ferrari 2010	PSM-CHMM
Apple logos	0.62/0.63	0.56/0.6	0.777/0.832	0.826/0.84
Bottles	0.501/0.51	0.38/0.42	0.798/0.816	0.91/0.93
Giraffes	0.36/0.426	0.328/0.34	0.399/0.445	0.833/0.854
Mugs	0.545/0.59	0.49/0.51	0.751/0.8	0.926/0.933
Swans	0.35/0.37	0.32/0.4	0.632/0.705	0.87/0.905
INRIA Horses	0.46/0.51	0.50/0.53	0.78/0.80	0.85/0.862

## Data Availability

Not applicable.

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
