# Peer review of "CHMM Object Detection Based on Polygon Contour Features by PSM"

_sensors, 2022, doi:10.3390/s22176556_

Round 1
Reviewer 1 Report
Please see the attached document for my comments and suggestions. I copy here the main questions I need an answer for:
line 77-84: This method looks very similar to what OpenCv FindContours does. How is this different? It may be best to show the results of this method versus known ones (like the OpenCV method) to highlight the differences. Also, add the computation time to get each resulting image.
line 101-109: What is j? You computed sharpness in equation 2, so what is this new formula? Do you mean that the sharpness of the contour is calculated as the maximum sharpness of the points in it?
line 112-118: Given the simplicity of the image, it's hard to imagine any contour algorithm failing in identifying the corner points. Again, can you show the differences in the same image of different algorithms versus your own?
line 129: It is no wonder that a curved line can be approximated by a certain number of points and the mathematical equation of the approximation can be found according to the polynomial degree. The more points, the more accurate the approximation is and the more computational time is required. However, this is a known theme and for example spline approximation usually is the best choice for most problems. Why you didn't implement it? This is basic knowledge. Moreover, your method is iterative, meaning that loops are involved and the computational time increases. Please comment on this matter.
line 245 (figure 14): Some are not perfectly enclosed. Is this an error, could it be improved...? The bounding box is found by the HMM starting from the contour approximation, is this correct? I would add a second figure showing the same picture analyzed by the different algorithms to provide a better visual comparison of their effectiveness to detect correct ROIs

Author Response
Q1: line 77-84: This method looks very similar to what OpenCv FindContours does. How is this different? It may be best to show the results of this method versus known ones (like the OpenCV method) to highlight the differences. Also, add the computation time to get each resulting image.
Reply: PSM algorithm has the some characters. Firstly,the corner detection algorithm based on sharpness is used to determine the contour corner, and the contour is segmented by the corner, which reduces the dependence of the contour segment on the starting point. Secondly, the distance ratio and arc length ratio are used to replace the distance error as the iterative stop condition, which improves the robustness of the object scale variance, such as the size, rotation, etc. Finally, both the angle and length extracted on the approximate polygon as two features describe the contour shape.
The advantages of PSM are the smaller number of the features and the higher detection rate. The number of the features for contour of the object is much smaller than that by PSM, shown in Table.1. the detection rate comparisons are shown in Fig.13 and Table.2.
Q2: line 101-109: What is j? You computed sharpness in equation 2, so what is this new formula? Do you mean that the sharpness of the contour is calculated as the maximum sharpness of the points in it?
Reply: we added the means of symbol j after Eq.(3).
We revised the symbol of the sharpness of the contour in Eq.(3).
Q3: line 112-118: Given the simplicity of the image, it's hard to imagine any contour algorithm failing in identifying the corner points. Again, can you show the differences in the same image of different algorithms versus your own?
Reply: Here, we emphasize the basic principle of PSM algorithm. PSM algorithm is compared to SM in section 2.2 and 2.3.
Q4: line 129: It is no wonder that a curved line can be approximated by a certain number of points and the mathematical equation of the approximation can be found according to the polynomial degree. The more points, the more accurate the approximation is and the more computational time is required. However, this is a known theme and for example spline approximation usually is the best choice for most problems. Why you didn't implement it? This is basic knowledge. Moreover, your method is iterative, meaning that loops are involved and the computational time increases. Please comment on this matter.
Reply: we would like to propose PSM algorithm not only to approximate the object contour, but also to approximate the contour curve with different scales, shown in Fig.7.
Q5: line 245 (figure 14): Some are not perfectly enclosed. Is this an error, could it be improved...? The bounding box is found by the HMM starting from the contour approximation, is this correct? I would add a second figure showing the same picture analyzed by the different algorithms to provide a better visual comparison of their effectiveness to detect correct ROIs.
Reply: we recompleted the simulations of the proposed algorithm to obtain the figures, in the revised verion.

Reviewer 2 Report
1. Abstract: "since they have a strong coupling relationship..." clumsy sentence, please rewrite this.
2. line 90-91. Rewrite
3. the figure description in line 182 is confusing. Also, figure legends are needed if different scale variables are shown in one figure.
4. line 188, how to quantify the "higher degree of coincidence" of the curves? It's not significantly evident in Figure 12.
5. More details are needed to clarify the CHMM processes. A simple diagram with notations/equations is needed, instead of having equations embedded in the text.
6. rewrite line 213. "which must be larger than..."
Author Response
Q1. Abstract: "since they have a strong coupling relationship..." clumsy sentence, please rewrite this.
Reply: the sentence “ and affect each other along the contour order ……. Since they coupling relationship.”
We revises these sentences as “ they have a strong coupling relationship since they affect each other along the contour order relationship.” See Line 8 -9 in the revised version.
Q2. line 90-91. Rewrite
Reply: we revised the sentence. See Line 93-94 in the revised version
Q3. the figure description in line 182 is confusing. Also, figure legends are needed if different scale variables are shown in one figure.
Reply: we revised the sentence as “ the length Lk and the angle θk sequences are extracted by PSM algorithm and SM algorithm for the bat object with the three scales of 100%, 60% and 90°, respectively.”. see line 185-186 in the revised version.
we revised Fig. 12.
Q4. line 188, how to quantify the "higher degree of coincidence" of the curves? It's not significantly evident in Figure 12.
Reply: we revised the sentence as “The three curves for the bat with the three scales of 100%, 60% and 90° are much more consistent in Fig. 12(a) than those in Fig. 12(b). Also for Fig. 12(b) and Fig. 12(d).”
The character of consistent is shown by comparisons from Fig.12(a)(c) to Fig.12(b)(d), since the three curves in Fig.12(a) or (c) varies much more consistent than those in Fig.12(b) or (d).
See line 189-197 in the revised version.
Q5. More details are needed to clarify the CHMM processes. A simple diagram with notations/equations is needed, instead of having equations embedded in the text.
Reply: In this works, we would like to show our contributions on PSM algorithm. So, we suggest to briefly descript the basic theory of CHMM with the consideration of the length limation of the manuscript. Moreover, The basic theory of CHMM algorithm is introduced in [18] and [22] in our manuscript.
Q6. rewrite line 213. "which must be larger than..."
Reply: we revise this sentence in line 219 of the revised version.
Round 2
Reviewer 1 Report
After the modifications performed by the authors to improve the manuscript, I think it is now ready for publication.
Reviewer 2 Report
The authors have managed to address all my comments satisfactorily. I'd like to thank the authors for their consideration and efforts.